# Longitudinal Evaluation of the Detection Potential of Serum Oligoelements Cu, Se and Zn for the Diagnosis of Alzheimer’s Disease in the 3xTg-AD Animal Model

**DOI:** 10.3390/ijms26083657

**Published:** 2025-04-12

**Authors:** Olivia F. M. Dias, Nicole M. E. Valle, Javier B. Mamani, Cicero J. S. Costa, Arielly H. Alves, Fernando A. Oliveira, Gabriel N. A. Rego, Marta C. S. Galanciak, Keithy Felix, Mariana P. Nucci, Lionel F. Gamarra

**Affiliations:** 1Hospital Israelita Albert Einstein, São Paulo 05652-000, SP, Brazil; olivia_fmd@hotmail.com (O.F.M.D.); nicolemev@gmail.com (N.M.E.V.); javierbm@einstein.br (J.B.M.); costacjs@gmail.com (C.J.S.C.); ariellydahora1997@gmail.com (A.H.A.); fernando.ao@einstein.br (F.A.O.); gnery.biomedicina@gmail.com (G.N.A.R.); marta.caetano.2004@gmail.com (M.C.S.G.); keithyflx@gmail.com (K.F.); 2LIM44—Hospital das Clínicas da Faculdade Medicina da Universidade de São Paulo, São Paulo 05403-000, SP, Brazil; mariana.nucci@hc.fm.usp.br

**Keywords:** Alzheimer’s disease, ICP-MS, biomarkers, oligoelements, animal model

## Abstract

Alzheimer’s disease (AD) is a progressive neurodegenerative disorder characterized by the accumulation of β-amyloid (Aβ) and hyperphosphorylated tau, leading to neuroinflammation, oxidative stress, and neuronal death. Early detection of AD remains a challenge, as clinical manifestations only emerge in the advanced stages, limiting therapeutic interventions. Minimally invasive biomarkers are essential for early identification and monitoring of disease progression. This study aims to evaluate the sensitivity of the relationship between serum oligoelement levels as biomarkers and the monitoring of AD progression in the 3xTg-AD model. Transgenic 3xTg-AD mice and C57BL/6 controls were evaluated over 12 months through serum oligoelement quantification using inductively coupled plasma mass spectrometry (ICP-MS), Aβ deposition via immunohistochemistry, and cognitive assessments using memory tests (Morris water maze and novel object recognition test), as well as spontaneous locomotion analysis using the open field test. The results demonstrated that oligoelements (copper, zinc, and selenium) were sensitive in detecting alterations in the AD group, preceding cognitive and motor deficits. Immunohistochemistry was performed for qualitative purposes, confirming the presence of β-amyloid in the CNS of transgenic animals. Up to the third month, labeling was moderate and restricted to neuronal cell bodies; from the fifth month onward, evident extracellular deposits emerged. Behavioral assessment indicated impairments in spatial and episodic memory, as well as altered locomotor patterns in AD mice. These findings reinforce that oligoelement variations may be associated with neurodegenerative processes, including oxidative stress and synaptic dysfunction. Thus, oligoelement analysis emerges as a promising approach for the early diagnosis of AD and the monitoring of disease progression, potentially contributing to the development of new therapeutic strategies.

## 1. Introduction

Alzheimer’s disease (AD) is the leading cause of dementia, accounting for approximately 60–70% of global cases, and represents one of the greatest public health challenges due to population aging [1,2]. Currently, it is estimated that 55 million people live with the disease, a number that could exceed 139 million by 2050 [3,4]. In addition to its social impact and effects on the quality of life of patients and caregivers, AD imposes a growing economic burden, with costs estimated at USD 1.3 trillion in 2019, potentially surpassing USD 2.8 trillion by 2030 if more effective therapeutic and diagnostic strategies are not implemented [1,5].

AD is a progressive neurodegenerative disease characterized by the accumulation of β-amyloid (Aβ) senile plaques and hyperphosphorylated Tau neurofibrillary tangles, leading to synaptic dysfunction, neuroinflammation, and neuronal death [6,7]. Evidence shows that these neuropathological changes emerge decades before the onset of clinical symptoms, making early diagnosis difficult and reducing the chances of effective intervention [8,9]. Diagnosing AD can be challenging, as clinical symptoms often take up to a decade to manifest, suggesting that the pathological changes associated with AD begin long before the appearance of visible cognitive signs [10].

Currently, the diagnosis of AD combines clinical evaluation, neuropsychological tests, neuroimaging exams, and biochemical biomarkers [11]. The detection of phosphorylated Tau forms (p-Tau181, p-Tau217, and p-Tau231) has shown a significant correlation with findings in cerebrospinal fluid and positron emission tomography (PET) [11]. Neurofilament light chain has also been investigated as a potential biomarker of neurodegeneration, although its specificity for AD is still debated [12]. However, despite advances in diagnosis, there is still no ideal biomarker that is widely accessible and reliable for early detection, driving the search for new strategies [13].

Among emerging approaches, the profile of oligoelements has been widely investigated in AD, considering their importance in brain homeostasis, redox metabolism, and protein aggregation. Imbalances in copper (Cu), zinc (Zn), and selenium (Se) levels have been associated with disease progression and increased oxidative damage to neuronal tissue [14,15]. Copper is essential for various neuronal functions, but its excess can induce cellular toxicity and promote Aβ aggregation [15]. Zinc, in turn, plays a role in synaptic modulation and antioxidant defense, but its altered concentrations may interfere with the structural stability of proteins involved in AD, impacting neuroplasticity [14]. Selenium, due to its antioxidant properties, plays a neuroprotective role by regulating lipid peroxidation and protecting neurons from the exacerbated oxidative stress observed in AD [16].

Inductively coupled plasma mass spectrometry (ICP-MS) has been widely used to quantify oligoelements in serum and brain tissue, allowing the correlation between their alterations and disease progression [17]. Experimental models, such as 3xTg-AD, have shown significant variations in the levels of these elements throughout the evolution of AD, highlighting their relevance for understanding pathological mechanisms and for the development of promising biomarkers [18,19].

The 3xTg-AD model is a widely used strain of AD, containing three familial mutations (APP Swedish, MAPT P301L, and PSEN1 M146V), and the translations of the overexpressed transgenes are restricted to the CNS, mostly the hippocampus and cortex. These mice display a progressive deposition of Aβ, appearing around 6 months of age in the frontal cortex. As for neurofibrillary tangles, the changes can be noticed by 12 to 15 months in the hippocampus [20,21]. Cognitive impairment can be noticed as early as 4 months of age, first manifesting as a retention deficit, and by 6 months, the learning and memory deficits are clear in this model [22,23].

Thus, this study aims to evaluate the relationship between serum oligoelements levels and AD progression in the 3xTg-AD model, using ICP-MS as an analytical tool. Identifying specific patterns may contribute to the development of accessible and non-invasive blood biomarkers, aiding in both the early diagnosis and the monitoring of Alzheimer’s disease.

## 2. Results

### 2.1. Behavior Evaluation

#### 2.1.1. Spontaneous Locomotion in the Open Field Test—Actimeter

The analysis of fast-frequency horizontal movements (F-MOV, Figure 1A) showed that both groups exhibited a decline in movement frequency during the first three months, with a more pronounced decrease in the control group (gray boxplot), which also had a higher mean compared to the AD group (red boxplot). However, from the fifth month onward, this pattern reversed, with the AD group maintaining a higher mean movement frequency than the C group. ANOVA revealed a significant effect for time and a time × group interaction (*p* < 0.001), while the group effect alone was not significant (*p* = 0.686). In the individual time-point analysis, group comparisons using the unpaired *t*-test showed significant differences (*p* < 0.05) at months 2, 3, 6, 7, and 9, as shown in Table 1.

In the analysis of slow-frequency horizontal movements (S-MOV, Figure 1B), the initial decline in movement frequency was less pronounced and similar between groups during the first three months. However, in the following months, the AD group exhibited a significantly higher frequency compared to the C group. ANOVA revealed significant effects for time and group (*p* < 0.001), as well as a significant time × group interaction (*p* = 0.029). In the unpaired *t*-test between groups at each time point, significant differences were observed from the 6th to the 11th month (Table 1).

The fast rearing movements (F-REA, Figure 1C) initially showed a higher frequency in the control group compared to the AD group during the first three months. However, this pattern reversed in the subsequent months. Unlike horizontal movements, the initial decline in frequency was less pronounced, suggesting a different progression pattern. ANOVA revealed a significant effect for time and a time × group interaction (*p* < 0.001), while the group effect alone was not significant (*p* = 0.945). In the unpaired *t*-test analysis at each time point, significant between-group differences (*p* < 0.01) were observed at months 2, 3, 4, and 9, as shown in Table 1.

In the slow rearing movement pattern (S-REA, Figure 1D), the frequency remained similar between groups, with a slight dominance of the control group during the first three months. However, from that point onward, the AD group exhibited consistently higher movement frequency. ANOVA revealed a significant group effect (*p* < 0.001), while time and time × group interaction were not statistically significant (*p* = 0.157 and *p* = 0.100, respectively). In the monthly group comparisons unpaired *t*-test, a significant difference was observed only at month 6 (*p* = 0.006), as shown in Table 1.

#### 2.1.2. Morris Water Maze Evaluation

The Morris water maze test, represented in Figure 2A, was evaluated using the latency variable (Figure 2B–F), which corresponds to the time the animal took to reach the platform, and the velocity variable (Figure 2G).

In the latency assessment over 12 months (Figure 2F), the control group (C), represented by gray boxplots, showed a progressive reduction in latency until the 5th month, followed by a plateau, indicating learning and adaptation to the task, as the animals required less time to reach the target. In contrast, the AD group (red boxplots) exhibited an initial reduction in latency during the first two months, followed by a sharp increase in the 4th month, which remained elevated for the rest of the 12-month period compared to the C group. This pattern likely reflects the pathophysiological impact of AD, with learning and memory impairments affecting the animals’ ability to remember the platform’s location, leading to longer search times.

ANOVA revealed a significant group effect (*p* < 0.001), while time and time × group interaction were not statistically significant (*p* = 0.176 and *p* = 0.338, respectively). In pairwise comparisons, significant differences between the groups were observed only at the 4th month and from the 7th month onward, as indicated in Table 2.

In the velocity analysis (Figure 2G), it was observed that in both groups, velocity values were higher during the first four months compared to the subsequent months. The AD group exhibited a higher average velocity than the control group in the 3rd and 4th months. However, from the 5th month onward, the control group maintained a higher average velocity than the AD group, except for the last two months, where the values became more similar. ANOVA revealed a significant group effect (*p* < 0.001), while time and time × group interaction were not statistically significant (*p* = 0.735 and *p* = 0.657, respectively).

In the between-group unpaired *t*-test analysis, conducted month by month, no significant differences were observed. Therefore, while velocity varied significantly over time, there was no significant difference between the groups, as both exhibited a similar pattern throughout the study period.

#### 2.1.3. Assessment of Novel Object Recognition

The values presented in the analysis were considered in the test phase, after training (Figure 2A,B), and the data were analyzed using the heat map, tracking the exploration pattern (Figure 2J–M). In the assessment of the recognition index of novel objects relative to the familiar object over 12 months (Figure 2N), the AD group presented higher scores (>50%, indicating greater preference for the novel object and high recognition of the familiar object) than the C group only during the first two months. From the 4th month onwards, there was a sharp decline (<50%, indicating greater exploration of the familiar object), followed by a gradual recovery, although the scores remained consistently close to 50% (indicating equal exploration) compared to the C group [24,25]. ANOVA analysis revealed a significant effect for time (*p* = 0.005), group (*p* < 0.001), and interaction (*p* = 0.002).

The unpaired *t*-test between groups revealed statistically significant differences at the following time points: 4th month (*p* = 0.007), 5th month (*p* < 0.001), 7th month (*p* = 0.016), 10th month (*p* = 0.014), and 12th month (*p* = 0.013). These results can be observed in the boxplot differences (red vs. gray) and in Table 3.

The findings are consistent with the literature, as this AD model only begins to exhibit behavioral deficits after the fourth month of life. Additionally, the progressive decline in recognition index scores in the AD group, compared to the control, may be linked to disease progression, resulting in lower scores over time, as demonstrated in this study.

In the analysis of average exploration speed over time (Figure 2O), the AD group showed a lower average than the control group up to the 5th month, after which this pattern reversed. ANOVA revealed a significant effect for time and a time × group interaction (*p* < 0.001), while the group effect alone was not significant (*p* = 0.174).

In the between-group unpaired *t*-test at each time point, significant differences in average speed were observed early on (2nd and 3rd months, *p* < 0.01) and later at the 10th month (*p* = 0.002), as shown in Table 3.

### 2.2. Evaluation of Oligoelements Levels in Serum Samples by ICP-MS

#### 2.2.1. Calibration Curve for Oligoelement Quantification in Serum Samples

The calibration curves for the reference standards of Zn, Se, and Cu were established using the isotopes ^64^Zn, ^78^Se, and ^63^Cu, respectively. The Pearson correlation coefficients (R) obtained were 0.9947 for Zn, 0.9997 for Se, and 0.9982 for Cu, indicating a high correlation between the experimental data and the calibration curves.

The LOD were 0.0101 ppb for Zn and Se, and 0.0132 ppb for Cu, while the LOQ were 0.0340 ppb for Zn, 0.0333 ppb for Se, and 0.0436 ppb for Cu.

The BEC was 0.0374 ppb for Zn, 0.0200 ppb for Se, and 0.0988 ppb for Cu, indicating low interference levels. The R-value for Zn (0.9947) demonstrates a strong correlation, ensuring its quantification with accuracy comparable to the other elements analyzed.

Thus, the results obtained for Se, Cu, and Zn can be considered reliable and robust for the analysis of oligoelements in serum samples.

#### 2.2.2. Quantification of Oligoelements in Serum Samples Using ICP-MS Technique

The oligoelement Cu exhibited a slight increase in concentration over 12 months in both groups, with higher averages in the AD group, particularly after the 3rd month, as shown in Figure 3A (red boxplot) and Table 4. ANOVA analysis revealed a significant group effect (*p* < 0.001), while the time effect (*p* = 0.210) and time × group interaction (*p* = 0.682) were not significant.

In the unpaired *t*-test between groups at each time point, a significant difference (*p* < 0.05) was observed in most months, except for the 2nd, 5th, and 8th months (*p* > 0.05), as detailed in Table 4.

For the oligoelement Se, a linear decline in concentration over 12 months was observed in both groups, with lower averages in the AD group, especially after the third month, as shown in Figure 3B (red boxplot) and Table 4. ANOVA analysis revealed a significant effect for time and group (*p* < 0.001), while the interaction effect was not significant (*p* = 0.121).

In the unpaired *t*-test between groups at each time point, significant differences were observed in almost all months, except for month 2 (*p* = 0.895), as detailed in Table 4.

A similar pattern to selenium was observed for the oligoelement Zn, with a decline in concentration over 12 months in both groups, though more pronounced in the AD group, as shown in Figure 3C (red boxplot). In the control group, the decline was less steep, and ANOVA analysis revealed a significant effect for time, group, and interaction (*p* < 0.001). In the between-group unpaired *t*-test at each time point, significant differences were observed in almost all months, except for the 2nd and 4th months (*p* = 0.890 and *p* = 0.080, respectively), as detailed in Table 4.

In the analysis of the correlation between the main behavioral parameters and the quantification of oligoelements over time (Figure 4), we observed that the early pathological changes, especially those related to memory, showed a significant correlation between latency and selenium levels (r = −0.658, *p* = 0.020; Figure 4A). During an intermediate stage of disease progression, selenium also showed a significant correlation with the recognition index, another important memory parameter (r = 0.648, *p* = 0.049), as shown in Figure 4B. In the later stage, this correlation with selenium became evident again, this time associated with fast horizontal movement (r = −0.585, *p* = 0.046; Figure 4D), while maintaining its association with the recognition index (r = 0.762, *p* = 0.037; Figure 4F).

Additionally, zinc demonstrated sensitivity in its correlation with slow horizontal movement (r = 0.594, *p* = 0.046), as shown in Figure 4C, and latency (r = −0.587, *p* = 0.049; Figure 4E), suggesting a possible involvement of this oligoelement in motor and cognitive functions during more advanced stages. On the other hand, copper was the only trace element that did not show statistically significant correlations over time with the variables analyzed.

### 2.3. Immunohistochemical Confirmation of Aβ Peptide Presence in the CNS

Immunohistochemistry confirmed the presence of β-amyloid deposits in the central nervous system, revealing intracellular immunolabeling in the cell bodies of pyramidal neurons in the cerebral cortex and their dendritic processes. Additionally, irregular extracellular deposits, consistent with β-amyloid plaques, were identified in both gray and white matter. These deposits, indicated by arrows in Figure 5, reflect the presence of β-amyloid in the evaluated transgenic animals.

In the early months, immunolabeling was observed primarily in neuronal bodies, with limited extracellular deposits. At the 2nd month (Figure 5A1–A3), neuronal immunolabeling was evident, but extracellular deposits were less apparent. At the 3rd month (Figure 5B1–B3), the pattern of immunolabeling appeared similar, suggesting initial stages of β-amyloid deposition.

From the 5th month onward (Figure 5C1–C3), intracellular labeling became more distinct, with clearly visible extracellular deposits. In the 6th and 7th months (Figure 5D1–D3,E1–E3), extracellular deposits were more noticeable, and neuronal immunolabeling remained evident, displaying both multifocal and diffuse patterns.

Between the 8th and 12th months (Figure 5F1–J1,F3–J3), there was a widespread presence of extracellular deposits alongside consistent immunolabeling in pyramidal neurons, suggesting advanced stages of β-amyloid deposition in transgenic animals.

In the control group (C57BL/6), some immunolabeling was observed, likely reflecting the endogenous presence of β-amyloid precursor protein (APP). However, extracellular deposits similar to those found in the transgenic animals were not detected at any of the evaluated time points, from the 2nd to the 12th month. The representative images K1, K2, and K3, presented in Figure 5, correspond to the 12-month-old control animal and illustrate the immunolabeling pattern observed in this group.

The qualitative observations present in Figure 5 suggest that the most evident β-amyloid deposition began from the 5th month. The immunolabeling pattern, indicated by arrows, supports the qualitative assessment of β-amyloid accumulation and its potential contribution to functional and structural changes in the CNS in transgenic AD models.

In Figure 5, images labeled (A–K)1, (A–K)2, and (A–K)3 were captured at 100×, 200×, and 400× magnification, respectively.

## 3. Discussion

This study employed robust behavioral and structural tools to assess the progression of Alzheimer’s disease in an experimentally validated model, aiming to investigate the detection potential of serum oligoelements Cu, Se, and Zn for the diagnosis of Alzheimer’s disease. The results demonstrate that oligoelements were sensitive in the early detection of the disease, preceding the first changes in recognition memory, learning, spatial memory, spontaneous locomotion disorders, and the structural alterations identified by histological analysis. Immunohistochemistry was performed qualitatively to confirm the presence and progression of β-amyloid accumulation in the CNS of transgenic animals. The quantification of these oligoelements was performed using ICP-MS, a highly sensitive technique for detecting trace metals, ensuring the accuracy and reliability of the data obtained. These findings suggest that oligoelements may represent a viable option for early disease detection, in addition to offering a complementary method for monitoring and understanding the progression of AD, which is currently predominantly based on behavioral, structural, and functional changes.

The multi-elemental analysis of oligoelements Cu, Zn, and Se revealed distinct variation patterns over 12 months between the control and AD groups, using serum samples from 3xTg-AD transgenic animals. However, only in the second month of evaluation was there no significant difference observed between the groups, suggesting that disease progression influences the alteration in the homeostasis of these elements [15,26,27,28].

Cu showed a progressive increase in serum concentration over time, being significantly higher in the AD group in most of the analyzed months, a pattern also reported in other studies [27,28,29,30,31,32,33,34,35]. This increase may be related to dysfunction in copper transport due to altered ATP7B expression, resulting in elevated non-ceruloplasmin-bound copper [15]. Free copper has a high catalytic capacity for redox reactions, promoting the formation of reactive oxygen species, increasing oxidative stress, and contributing to the formation of Aβ aggregates [28,29]. The deposition of copper in Aβ plaques in brain tissue may also lead to a redistribution of the metal, contributing to its elevation in serum [33].

However, selenium, which plays a fundamental role in neuronal antioxidant defense as an essential cofactor of glutathione peroxidase, protecting cells against oxidative stress [27], showed a linear decline pattern over time in our study, with a more pronounced reduction in the AD group. This decline is similar to the results found in the literature [27,35,36,37,38,39,40]. Only in the second month were the differences between the groups not statistically significant, but as the disease progressed, the decline in Se concentration became evident. Its reduction in AD patients may be associated with the increased oxidative damage and neuronal loss observed in the disease [32]. Studies indicate that reduced selenium levels are correlated with cognitive decline and greater vulnerability to oxidative stress, which may accelerate neurodegeneration [30,40].

In the case of zinc, which plays an essential role in synaptic signaling and protein structure regulation, which is crucial for the stabilization of Aβ aggregates, our study also showed a declining pattern, with a more pronounced reduction in the AD group, as reported in the literature [27,28,30,33,36,41]. Its reduction may be associated with alterations in synaptic function and impaired neuronal plasticity [28,41]. In the second month, the differences between the groups were not statistically significant, but from the third month onward, the reduction in zinc levels in the AD group became more evident over time. Studies suggest that lower serum zinc levels are associated with increased oxidative stress and the exacerbation of neurodegeneration [26]. Additionally, the interaction of zinc with the Tau protein may influence the development of the neurofibrillary tangles characteristic of AD [29].

The results found in the analysis of oligoelements confirm the dysregulation of metal homeostasis in AD and reinforce their potential as early disease biomarkers and for monitoring disease progression. The increase in serum copper and the reduction in selenium and zinc levels may play fundamental roles in AD pathophysiology, contributing to oxidative and neuroinflammatory changes [27,33].

To confirm the pathophysiological progression of the disease, an immunohistochemical analysis was performed, revealing the presence of β-amyloid deposits in the central nervous system of 3xTg-AD transgenic mice. The staining showed intracellular immunoreactivity in the cell bodies of pyramidal neurons in the cerebral cortex and their dendritic processes, as well as irregular extracellular deposits consistent with β-amyloid plaques distributed in both gray and white matter. These qualitative findings are consistent with the literature [20,22,23,42], which describes the 3xTg-AD model as widely used for studying Alzheimer’s disease due to mutations in the APP, PS1, and Tau genes, which allow the formation of characteristic lesions such as Aβ deposits and hyperphosphorylated Tau neurofibrillary tangles [43]. Although positive staining was observed from three months of age, indicating early onset of peptide accumulation, no quantification of β-amyloid levels was performed, limiting the analysis to the confirmation of its presence in the evaluated tissues.

Among the most studied pathological events in Alzheimer’s disease, Aβ accumulation in the brain has been widely associated with the onset and progression of the disease. In this study, the positive staining observed in transgenic mice confirms the presence of the Aβ peptide in the central nervous system, in agreement with previously reported findings in the literature related to the neurodegenerative condition of AD [23,44,45]. In the early months, immunoreactivity was predominantly intracellular, localized in the cell bodies of pyramidal neurons, with extracellular deposits still barely noticeable. The formation of amyloid deposits follows a well-established temporal pattern, with initial appearance between 4 and 6 months of age and increased intensity between 9 and 12 months [46,47]. These findings were replicated in the present study, which showed moderate immunostaining in the early months, with a progressive increase in Aβ deposition over time. From the fifth month onward, extracellular staining became more evident, with more defined deposits, suggesting the gradual progression of the pathological process.

The impact of Aβ deposition in the CNS goes beyond plaque formation, affecting synaptic transmission and contributing to progressive cognitive deficits, as demonstrated by Jucker and Walker [48]. These findings reinforce the relationship between Aβ accumulation and synaptic dysfunction, a hallmark of AD. The analysis conducted by Mastrangelo et al. [44] further highlights that the progression of Aβ plaque burden is accompanied by other neuropathological changes, such as gliosis and the formation of tau tangles, which interact synergistically to worsen neuronal degeneration. The present study also identified that Aβ accumulation is more pronounced in critical regions such as the cortex and hippocampus, areas associated with memory and cognition.

Additionally, the interaction between Aβ and tau, commonly described in the literature, appears to play a critical role in facilitating disease progression [22,23,49,50]. Aβ accumulation precedes tau elevation in early stages, but in advanced phases, these two pathologies interact synergistically, exacerbating neuronal degeneration [44,47,51]. In the present study, immunohistochemical confirmation of β-amyloid presence in the central nervous system was performed, demonstrating its progressive accumulation over time. However, quantification of Aβ levels was not conducted, which represents a limitation. Future studies should include this quantitative analysis to deepen the understanding of the relationship between amyloid burden, functional deficits, and the progression of neurodegeneration.

As a complement to the assessment of disease progression, a behavioral evaluation of 3xTg-AD transgenic mice was conducted using three validated tests to investigate cognitive and motor deficits associated with AD. Open-field habituation (actimeter) was used to analyze spontaneous locomotion, while the Morris water maze assessed spatial memory and reference learning, both of which are highly dependent on hippocampal function. Additionally, episodic memory was evaluated using the novel object recognition test, a sensitive paradigm for detecting early deficits in recognition memory. These tests are widely applied in experimental AD models and allow for the correlation of behavioral changes with the underlying neurobiological mechanisms of disease progression [52,53,54].

The analysis of spontaneous locomotion results in the open field revealed significant differences between transgenic (AD) and control (C) mice. AD animals exhibited increased locomotor activity, both in vertical and horizontal movements, especially in the early months (2 to 6 months). This initial hyperactivity pattern suggests alterations in motor and dopaminergic control, which are often associated with the early progression of AD [53,55]. These findings are consistent with neuropathological data observed in the 3xTg-AD model, which show intracellular Aβ deposits around the third month of life, with evident progression after the fourth month. This correlation between neuropathological alterations and behavioral patterns reinforces the validity of the model for studying the early stages of the disease [56,57]. Rapid horizontal movements were more frequent in AD mice early on, indicating increased environmental exploration, possibly as an initial compensatory response to neuropathological changes. Studies suggest this heightened activity reflects neural compensation before more advanced motor and cognitive deficits emerge due to extracellular plaques and hyperphosphorylated Tau accumulation [53,58]. In contrast, AD mice exhibited relatively more frequent slow vertical and horizontal movements compared to controls, indicating a mix of anxiety and excessive exploration, often associated with hippocampal dysfunction [59].

In the Morris water maze, transgenic mice exhibited progressively more evident spatial memory deficits throughout the experiment. Although the latency to find the platform did not show significant differences in the early stages, this latency increased from the fourth month onward, coinciding with the neuropathological progression described in the model, such as the presence of extracellular plaques and hyperphosphorylated Tau in the hippocampus and cortex [53,56,60]. These findings reflect the progressive dysfunction of the hippocampus, a key area for spatial memory, which is frequently affected in the early stages of AD [20,61,62].

In the novel object recognition test, the recognition index was reduced in AD mice compared to controls, especially after the first months. Initially, AD animals could distinguish between the familiar and the new object, but this ability gradually declined, reflecting episodic memory deficits and synaptic plasticity alterations. These results align with neuropathological findings indicating the emergence of hyperphosphorylated Tau and its progression into neurofibrillary tangles from the sixth month onward, marking a progressive decline in cognitive capacity [53,58,63].

These findings confirm specific behavioral and cognitive changes in transgenic 3xTg-AD mice, reflecting the progressive characteristics of AD. The initial hyperactivity observed in association with intracellular Aβ deposits, combined with progressive deficits in spatial and episodic memory correlated with advancing Tau pathology, reinforces the relevance of this model for studying the underlying mechanisms of neurodegeneration and for investigating targeted therapeutic interventions.

However, this study presented some limitations, such as the lack of a complementary exploration of other oligoelements and the validation of these elements in cerebrospinal fluid samples in addition to serum. Although immunohistochemical confirmation of the presence of beta-amyloid (Aβ) peptide in the central nervous system was performed, a quantitative assessment of its levels was not carried out, which could have strengthened the correlation between Aβ deposition and changes in oligoelement levels. Furthermore, the absence of a broader approach to investigate the interactions of these elements with other biomolecules involved in the pathophysiology of Alzheimer’s disease limits a full understanding of the role they play in disease progression. Future studies should address these aspects to enhance the applicability of oligoelements as biomarkers for early diagnosis and disease monitoring.

## 4. Materials and Methods

### 4.1. Animals

All procedures followed the guidelines of the Ethics Committee on the Use of Animals at Hospital Israelita Albert Einstein (HIAE) under protocol CEUA 5446-22. The study used 57 male 3xTg-AD transgenic mice and 24 male C57BL/6 mice (control group), obtained from the Faculdade de Medicina, Universidade de São Paulo (FMUSP). The animals were housed in sterile cages under laminar airflow hoods at the Centro de Experimentação e Treinamento em Cirurgia (CETEC) animal facility of the Instituto Israelita de Ensino e Pesquisa Albert Einstein (IIEPAE), in a temperature-controlled room (22 ± 3 °C) with regulated humidity and a 12 h light/dark cycle. Pelleted commercial feed (Nuvilab CR-1, Nuvital, Quimtia, Brazil) and water were provided ad libitum.

### 4.2. Experimental Design

The experiment was conducted in three main phases, as detailed in Figure 6. In the first phase, behavioral assessments were performed in 3xTg-AD mice, which develop an AD-like pathology, using tests for spontaneous locomotion (open field test) from the 2nd to the 12th month and spatial memory (Morris water maze—MWM) and episodic memory (novel object recognition test) at months 2, 3, 5, 6, 7, 8, 10, and 12. In the second phase, serum oligoelement quantification was carried out using ICP-MS in serum samples from C57BL/6 control mice and 3xTg-AD mice, with monthly blood collections from the 2nd to the 12th month. In the third phase, Aβ peptide deposition analysis was performed by immunohistochemistry (IHC) in the brains of 33 3xTg-AD mice (*n* = 3 per month), collected at months 2, 3, 5, 6, 7, 8, 10, and 12. Additionally, brain samples from C57BL/6 control mice, which underwent blood collection and behavioral assessment, were also analyzed for comparison.

### 4.3. Behavioral Assessment

#### 4.3.1. Spontaneous Locomotion in the Open Field—Actimeter

Spontaneous locomotor activity was quantified using the Infrared Actimeter LE 8825 system (Actitrack, Panlab Harvard Apparatus, Barcelona, Spain), which consists of a bidimensional surface (70 × 70 cm^2^) surrounded by transparent walls 30 cm in height. The system includes infrared sensors that detect horizontal and vertical movements, allowing the analysis of locomotor activity. The four movement parameters used for comparison between groups and sessions were slow horizontal (S-MOV), fast horizontal (F-MOV), slow rearing (S-REA), and fast rearing (F-REA). Rearing movement can be considered as a vertical movement, reflecting exploratory behavior.

Animals were placed at the center of the arena and tracked for 5 min. Locomotor activity was recorded based on the interruption of light beams, distinguishing movement types. Data were processed using SEDACOM v2.0 software.

At the end of each session, animals were returned to their cages, and the equipment surface was cleaned with 5% alcohol. The frequency of S-MOV, F-MOV, S-REA, and F-REA movements was analyzed over 12 months in the control and AD model groups.

#### 4.3.2. Morris Water Maze

The Morris water maze tests were conducted in a 180 cm diameter circular pool with a transparent platform 1 cm below the water surface. The pool was filled with 37 °C water, opacified with 400 g of milk powder, and the room temperature was maintained at 21 °C.

In the acquisition phase, mice had 60 s to find the platform. Those that succeeded remained for 10 s, while the others were guided to it for the same duration. Mice were always released from the same quadrant (diagonally opposite to the platform) and underwent four trials per day for five consecutive days.

On day six, mice had 60 s to locate the platform without any visual cue. Those that failed were excluded from the study.

The analyzed parameters were escape latency (s) and average swimming speed (cm/s), processed using Noldus EthoVision XT (Version 14, Noldus Information Technology, Wageningen, The Netherlands).

#### 4.3.3. Novel Object Recognition

The novel object recognition test is used to assess cognition, particularly short-term memory, in murine models with central nervous system (CNS) disorders, based on the natural tendency of rodents to spend more time exploring a novel object than a familiar one. The task was conducted in a 70 × 70 cm open-field arena with a white floor and 30 cm high walls and consisted of three phases: in the habituation phase, animals freely explored the empty arena for 5 min; in the familiarization phase, they were exposed to two identical objects for 5 min; and in the test phase, one of the objects was replaced with a novel one, always positioned in the eastern quadrant, allowing recognition analysis.

The tests were recorded using EthoVision XT software (Version 14, Noldus Information Technology, Wageningen, The Netherlands) under low lighting conditions (20–40 lux) and conducted in the morning. Evaluations were performed at months 2, 3, 4, 5, 6, 7, 8, 10, and 12, enabling the assessment of performance progression over time.

The analyzed parameters included recognition index (%), average locomotion speed (cm/s), and total exploration time (s). Average speed was considered to evaluate possible locomotion effects on object exploration, while total exploration time reflected the interaction of the animals with both objects. The recognition index (RI) was calculated using the equation ${\;RI=T}_{N}/(T_{N}+T_{F)}\;\times \;100$, where *T_N_* represents the exploration time of the novel object and *T_F_* the exploration time of the familiar object. Values closer to 100% indicate a greater preference for the novel object, reflecting better recognition memory retention, whereas lower values may suggest cognitive impairments associated with the experimental model.

### 4.4. Determination of Oligoelements in Serum Samples by ICP-MS

#### 4.4.1. Blood Sample Collection and Pre-Preparation

Blood collection was performed under isoflurane anesthesia Isoforine^®^ (Cristália, Itapira, Brazil) using an acrylic induction chamber connected to an anesthesia system. A 2.5 mL/min O_2_ flow with 2.5% isoflurane saturation was administered. Once immobility and loss of postural reflexes were confirmed (approximately 2 min after induction), blood sampling was initiated. Blood was collected intravenously through the retro-orbital sinus, with 500 µL extracted monthly from each animal into a 1.5 mL microtube (Eppendorf^®^, São paulo, Brazil). The serum was separated by centrifugation at 3500 rpm for 15 min at 4 °C, followed by freezing at −80 °C for further analysis.

#### 4.4.2. Preparation of Samples for Quantitative Measurements by ICP-MS

For sample preparation, the serum samples were removed from storage at −80 °C, thawed at room temperature, and vortexed to ensure complete homogenization. The process began with the addition of 100 μL of each sample into a 1.5 mL conical-bottom polypropylene tube (Eppendorf^®^, São Paulo, Brazil), followed by the addition of 300 μL of HNO_3_ (15%) and 100 μL of H_2_O_2_. The tubes containing the samples were subjected to thermal treatment at 60 °C with agitation at 450 rpm using ThermoMixer C equipment (Eppendorf^®^, São Paulo, Brazil) until the samples became transparent. Finally, clear samples were obtained, indicating efficient digestion. The digested samples were stored at −20 °C until quantitative analysis by ICP-MS. For quantitative analysis, the samples were diluted 500X (five hundred times) in Milli-Q water with 2% HNO_3_.

#### 4.4.3. Quantitative Analysis of Oligoelements in Serum Samples by ICP-MS

The quantitative determination of Cu, Zn, and Se in blood serum samples was performed using an ICP-MS NexION 350X (PerkinElmer®, Waltham, Ma, USA), through the analysis of the nuclides ^63^Cu, ^64^Zn, and ^78^Se. The instrument is equipped with a nickel sample cone (W10336112, Perkin Elmer), nickel skimmer cone (W1026356, Perkin Elmer), and aluminum hyperskimmer cone (W1033995, Perkin Elmer); a concentric glass nebulizer; a 0.5 mm internal diameter probe; and a cyclonic spray chamber, ensuring high sensitivity and efficient aerosol generation.

Quantification was initiated by constructing calibration curves using certified standard solutions of Cu (N9300183, Perkin Elmer), Zn (N9303758, Perkin Elmer), and Se (N9303752, Perkin Elmer) at a concentration of 1000 ppm. These standards were diluted to final concentrations of 0.2, 0.5, 0.8, 1, and 2 ppb in Milli-Q water containing 2% HNO_3_, maintaining a final volume of 10 mL. Instrumental variation was monitored by preparing an additional set of standards at a known concentration of 0.8 ppb. Normal mouse serum (NS03L, Sigma-Aldrich®, St. Louis, MO, USA) was used as a reference standard for quantification validation and subjected to the same preparation process as the serum samples. Rhodium was incorporated as an internal standard to correct for spectral interferences, enhancing analytical accuracy. The limit of detection (LOD), limit of quantification (LOQ), and background equivalent concentration (BEC) of Cu, Zn, and Se were determined from the calibration curves using Syngistix software, version 4.0 (PerkinElmer®, Waltham, MA, USA).

Sample introduction and plasma conditions were optimized to ensure high signal stability and minimal oxide formation. The sample flow rate was automatically regulated, while the argon gas flow and torch position were adjusted daily for optimal performance. Quantification was carried out using the dynamic reaction cell (DRC) mode under fixed parameters, including 1600 W RF power, 18 L/min plasma flow, 1.0 L/min auxiliary flow, and 0.93 L/min nebulizer gas flow. Data acquisition consisted of 50 readings per replicate, with three measurements per sample and a dwell time of 50 ms per isotope. To assess method sensitivity, sample dilution was performed until reaching the detection limits for Cu, Zn, and Se, using the established calibration curves. Quantification was conducted in triplicate, and the intermediate value among the three measurements was used for the final analysis. By selecting the intermediate value, we aimed to enhance the stability of the measurements, ensuring greater reproducibility while reducing potential distortions caused by experimental variations. Spectral interferences, particularly isobaric and polyatomic interferences, were minimized by optimizing the DRC with methane as the reaction gas. The methane flow rate was precisely controlled to enhance the signal-to-noise ratio, improve plasma stability, and ensure consistent measurements in complex biological matrices.

### 4.5. Identification of Aβ Peptide Deposition in the CNS by Immunohistochemistry

The presence of total Aβ peptide was identified by IHC in genetically modified 3xTg-AD mice, a well-established model for Alzheimer’s disease. Brain samples were collected at 2, 3, 5, 6, 7, 8, 9, 10, 11, and 12 months and fixed in 10% paraformaldehyde solution at 4 °C for 24 h. Subsequently, the samples were transferred to a 30% sucrose solution and stored at 4 °C for approximately 48 h, or until complete tissue sinking was observed. Following this process, brains were embedded in optimal cutting temperature (OCT) compound and stored at −80 °C until sectioning. Tissue sections were obtained using a cryostat (Leica Biosystems, Nussloch, Germany) at a thickness of 15 µm.

The sections were air-dried at room temperature for 20 min, followed by immersion in pre-chilled ethanol for 10 min and rehydration in phosphate buffer (three washes of 10 min each). Endogenous peroxidase activity was blocked by incubation in 0.3% hydrogen peroxide. The sections were then incubated with the primary antibody (Beta Amyloid Polyclonal Antibody, 36–6900; Thermo Fisher Scientific®, Waltham, MA, USA) at a dilution of 1:200 in 1% bovine serum albumin (BSA) and 0.3% Triton X-100 solution. Signal detection was performed using diaminobenzidine (DAB), with sections immersed in DAB solution for 1 min and 30 s, followed by three washes of 5 min each. Counterstaining was performed with Harris hematoxylin for 5 min, followed by graded ethanol dehydration (70%, 90%, and 100%), immersion in xylene, and coverslipping with Permount™ (Thermo Fisher Scientific®, Waltham, MA, USA).

Immunolabeling was evaluated in specific brain regions, including the basomedial amygdala (−1.70 to −1.22 mm post-Bregma), cortex (−2.06 to −1.70 mm post-Bregma), CA1 region of the hippocampus (−2.06 to −1.70 mm post-Bregma), and subiculum (−3.08 to −2.80 mm post-Bregma).

### 4.6. Statistcal Analysis

The data were presented as mean and standard deviation for each analyzed group. Normality and variance were assessed using the Shapiro–Wilk test and Levene’s test, respectively, to determine the appropriate statistical test. Considering the factors, time, and group, an ANOVA was performed, while cross-sectional comparisons at each session were analyzed using an unpaired *t*-test. The most relevant behavioral data and the quantification of oligoelements were correlated using either Spearman’s or Pearson’s test, depending on the data distribution. All statistical analyses were conducted using JASP (version 0.16.4) and Origin 9.1 (OriginLab, Northampton, MA, USA). Results were considered statistically significant for *p*-values < 0.05.

## 5. Conclusions

This study demonstrates that oligoelements have potential as promising candidates for the early diagnosis of Alzheimer’s disease, as their alterations precede the cognitive and structural deficits observed in the 3xTg-AD model. Elevated copper levels were associated with increased oxidative stress, while the reduction of zinc and selenium was correlated with progressive neurodegeneration. Immunohistochemical analysis confirmed the accumulation of β-amyloid peptide and its relationship with synaptic dysfunction. Behavioral changes, especially in the spatial and episodic memory domains, reinforce disease progression and its association with Aβ deposition and hyperphosphorylated Tau. The ICP-MS technique proved effective in detecting these variations, consolidating its use as a promising tool for the diagnosis and monitoring of the disease. Additionally, the correlation analysis between behavioral parameters and oligoelement levels showed that selenium and zinc are significantly associated with cognitive and motor functions at different stages of disease progression, reinforcing their potential as functional markers. These findings suggest that the analysis of oligoelements may contribute to the development of earlier and more precise strategies for the detection and monitoring of disease progression, as well as understanding of the mechanisms involved.

## Figures and Tables

**Figure 1 ijms-26-03657-f001:**
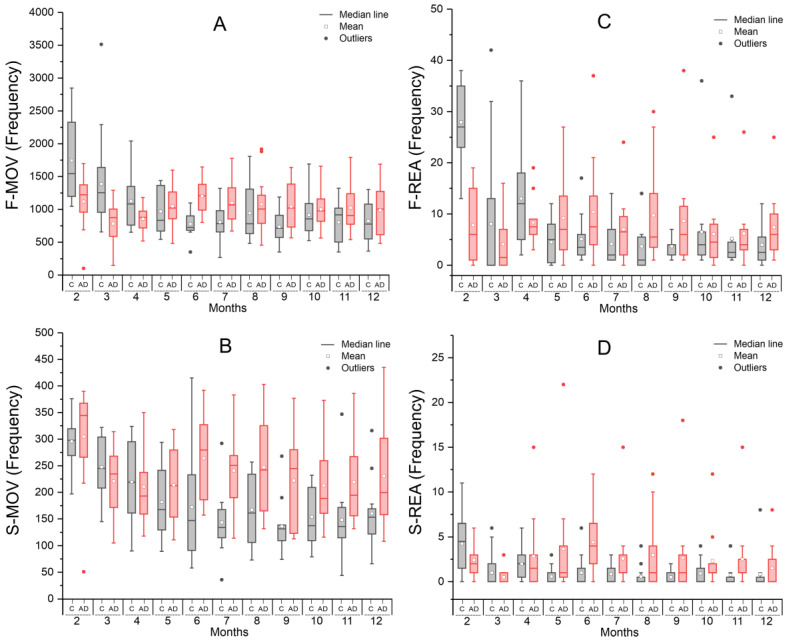
Analysis of spontaneous locomotion in the open field test—actimeter, based on movement frequency over 12 months for the control group (gray) and the Alzheimer’s disease group (red): (**A**) frequency of fast horizontal movements; (**B**) frequency of slow horizontal movements; (**C**) frequency of fast rearing movements; and (**D**) frequency of slow rearing movements.

**Figure 2 ijms-26-03657-f002:**
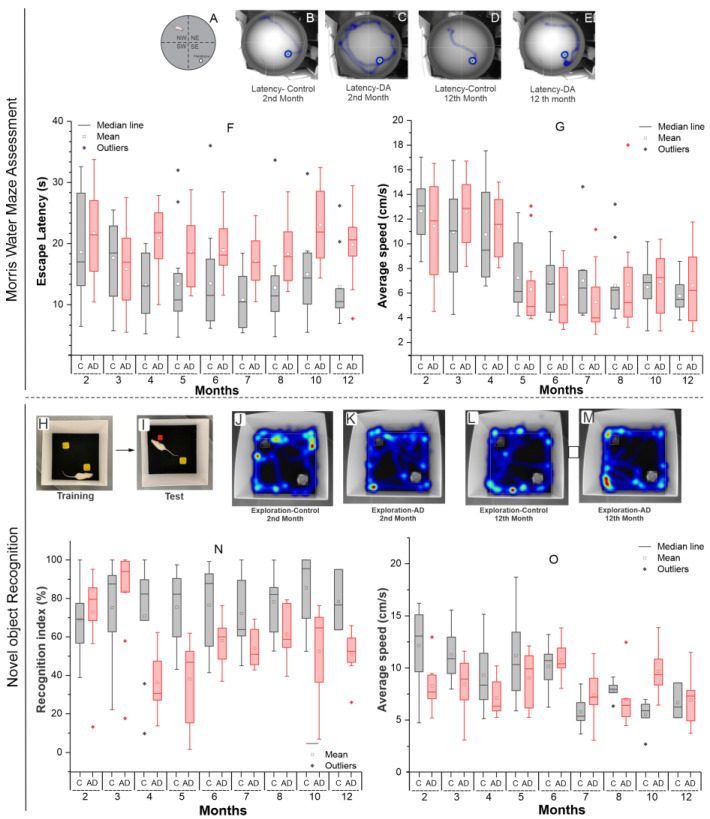
Assessment of recent and long-term memory, as well as learning, through the (**A**–**E**) Morris water maze Test, including (**F**) analysis of the latency variable (time in seconds for the animal to reach the platform by swimming) and (**G**) the average speed of the control and Alzheimer’s disease groups over 12 months. Additionally, evaluation was performed using the (**H**–**O**) novel object recognition test in the test phase after training, where (**H**) refers to the phase with two yellow familiar objects, and (**I**) introduces a red object as the novel one. The data were analyzed using the heat map, tracking the exploration pattern (**J**–**M**), considering (**N**) the recognition index and (**O**) the average speed.

**Figure 3 ijms-26-03657-f003:**
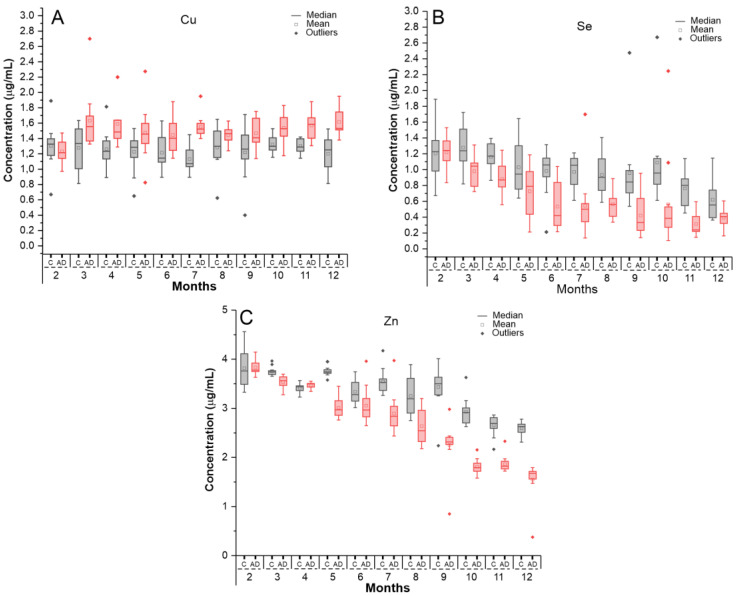
Analysis of oligoelements based on the concentrations of (**A**) copper (Cu), (**B**) selenium (Se), and (**C**) zinc (Zn) over 12 months in the Alzheimer’s disease group (red) and control group (gray).

**Figure 4 ijms-26-03657-f004:**
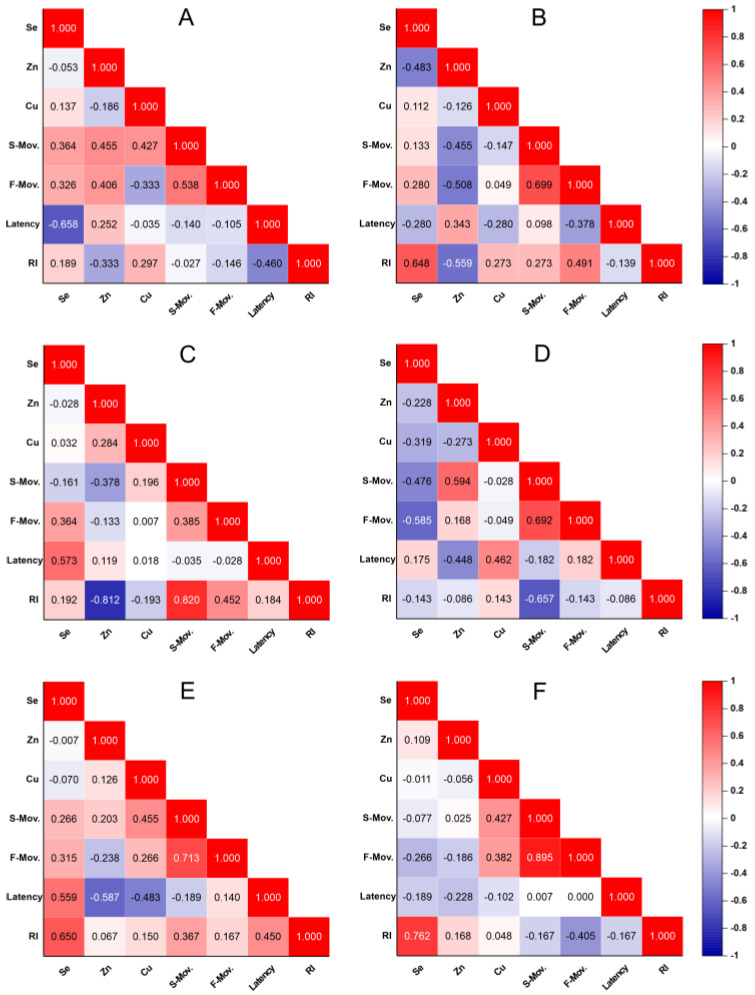
Correlation analyses between the quantification of oligoelements and key behavioral parameters across different time points. The correlation matrices refer to: (**A**) third month, (**B**) fifth month, (**C**) seventh month, (**D**) eighth month, (**E**) tenth month, and (**F**) twelfth month. Color gradients represent the strength and direction of the Spearman’s rank correlation coefficients (ρ), ranging from −1 (blue, strong negative correlation) to +1 (red, strong positive correlation). White-colored values indicate statistically significant correlations (*p* < 0.05). Behavioral parameters include slow horizontal movement (S. Mov.), fast horizontal movement (F. Mov.), latency, and recognition index (RI). The oligoelements analyzed were selenium (Se), zinc (Zn), and copper (Cu).

**Figure 5 ijms-26-03657-f005:**
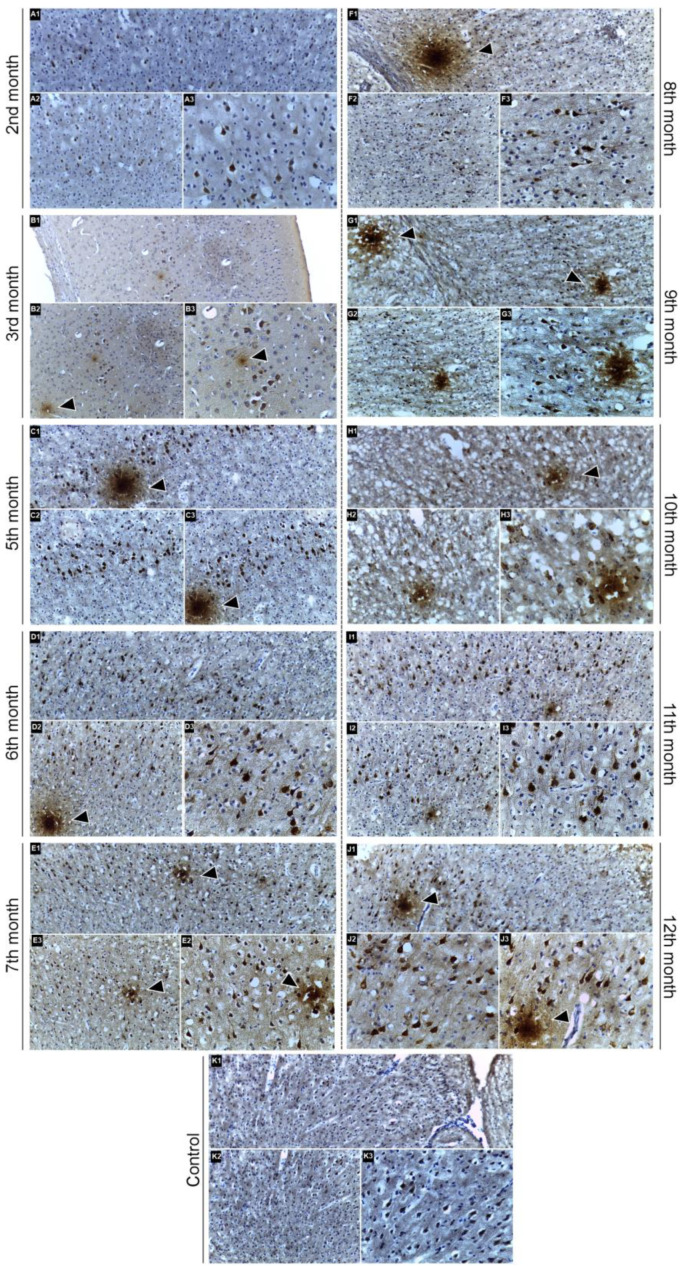
Immunolabeling of β-amyloid in the cerebral cortex of 3xTg-AD transgenic animals at different experimental time points. Intracytoplasmic immunolabeling is observed in the cell bodies of pyramidal neurons and their dendritic processes, while extracellular deposits (indicated by arrows) are irregularly distributed in the gray and white matter. Panels (**A1**–**A3**) represent the 2nd month, (**B1**–**B3**) the 3rd month, (**C1**–**C3**) the 5th month, (**D1**–**D3**) the 6th month, (**E1–E3**) the 7th month, (**F1–F3**) the 8th month, (**G1–G3**) the 9th month, (**H1–H3**) the 10th month, (**I1–I3**) the 11th month and panels (**J1**–**J3**), which represent the 12th month. The control group (C57BL/6) showed no intracellular immunolabeling or extracellular deposits at any evaluated time points. To represent these consistent control characteristics, representative images labeled (**K1**–**K3**) were included.

**Figure 6 ijms-26-03657-f006:**
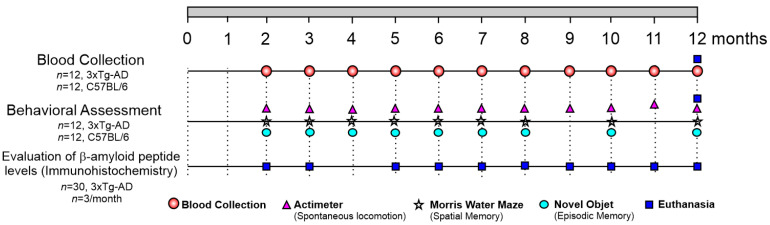
Experimental design for the evaluation of biomarkers in the 3xTg-AD Alzheimer’s disease model. The study was divided into three phases: behavioral assessments, including spontaneous locomotion (months 2–12) and memory tests (months 2, 3, 5, 6, 7, 8, 10, and 12); serum oligoelement quantification by ICP-MS (months 2–12); and Aβ peptide deposition analysis by immunohistochemistry (months 2, 3, 5, 6, 7, 8, 10, and 12). The colored markings represent each time point analyzed, not the number of animals assessed at each time point.

**Table 1 ijms-26-03657-t001:** Descriptive and statistical analysis of actimeter data for Alzheimer’s disease and control groups: comparison of horizontal and vertical movements at different frequencies (Fast and Slow).

Time (Months)	Groups		Statistic *p*-Value
	Control	Alzheimer Disease	
	Mean ± SD	Mean ± SD	
F-MOV			
2	1744.58 ± 658.25	1119.92 ± 413.06	**0.011**
3	1387.16 ± 671.87	783.33 ± 335.06	**0.007**
4	1125.08 ± 413.62	856.08 ± 198.87	0.055
5	968.58 ± 344.11	1048.08 ± 334.43	0.572
6	777.83 ± 199.82	1204.00 ± 279.25	**<0.001**
7	806.83 ± 268.97	1100.92 ± 322.09	**0.024**
8	944.50 ± 423.63	1076.58 ± 445.10	0.464
9	736.25 ± 238.16	1044.42 ± 367.28	**0.023**
10	914.58 ± 343.90	1000.33 ± 292.59	0.517
11	802.67 ± 316.87	1025.75 ± 382.31	0.134
12	824.83 ± 321.93	986.42 ± 391.42	0.281
S-MOV			
2	295.67 ± 47.53	304.667 ± 96.51	0.775
3	247.53 ± 54.71	220.667 ± 65.23	0.226
4	219.33 ± 76.05	210.667 ± 71.24	0.776
5	181.58 ± 66.33	213.58 ± 69.22	0.260
6	172.42 ± 109.04	264.08 ± 84.29	**0.031**
7	143.33 ± 61.01	240.75 ± 80.81	**0.003**
8	167.25 ± 70.15	247.50 ± 93.33	**0.026**
9	136.42 ± 50.41	222.25 ± 91.55	**0.009**
10	153.83 ± 54.89	212.83 ± 78.16	**0.044**
11	148.08 ± 74.76	218.92 ± 86.68	**0.043**
12	160.17 ± 64.96	230.33 ± 99.07	0.052
F-REA			
2	27.917 ± 7.73	7.833 ± 7.29	**0.005**
3	8.053 ± 12.84	4.083 ± 5.16	**0.001**
4	13.000 ± 9.91	8.333 ± 4.52	**0.014**
5	4.500 ± 4.14	9.250 ± 8.51	0.754
6	5.083 ± 4.50	10.417 ± 10.21	0.095
7	4.167 ± 4.15	7.000 ± 6.55	0.088
8	3.667 ± 5.24	9.750 ± 9.77	0.967
9	3.667 ± 1.87	8.667 ± 10.23	**0.007**
10	6.500 ± 9.59	6.083 ± 6.71	0.297
11	5.167 ± 8.88	6.167 ± 6.73	0.475
12	4.000 ± 3.93	7.417 ± 6.64	0.113
S-REA			
2	4.25 ± 3.42	2.42 ± 1.98	0.122
3	1.00 ± 1.83	0.50 ± 0.90	0.387
4	2.00 ± 1.86	2.83 ± 4.32	0.546
5	0.58 ± 0.99	3.67 ± 6.17	0.101
6	1.00 ± 1.86	4.42 ± 3.45	**0.006**
7	0.83 ± 1.19	2.58 ± 4.12	0.172
8	0.58 ± 1.24	3.00 ± 4.07	0.068
9	0.50 ± 0.79	2.67 ± 5.05	0.156
10	0.92 ± 1.38	2.33 ± 3.31	0.185
11	0.75 ± 1.54	2.50 ± 4.10	0.180
12	0.83 ± 2.29	1.50 ± 2.47	0.500

Note: The statistics presented are from the *t*-test, comparing groups at each time point, with *p*-values < 0.05 in bold.

**Table 2 ijms-26-03657-t002:** Descriptive and statistical analysis of Morris water maze data for Alzheimer’s disease and control groups: comparison of escape latency and average speed.

Time (Months)	Groups				Statistics *p*-Value
		Control		Alzheimer Disease	
	N	Mean ± SD	N	Mean ± SD	
Escape latency (s)					
2	9	18.61 ± 9.32	7	21.705 ± 7.561	0.487
3	11	17.66 ± 6.62	12	15.845 ± 6.827	0.526
4	8	13.19 ± 5.51	10	20.915 ± 5.554	**0.010**
5	12	13.44 ± 8.12	12	18.724 ± 5.844	0.081
6	11	13.46 ± 8.88	10	18.889 ± 4.825	0.103
7	8	10.82 ± 4.85	12	17.073 ± 4.194	**0.007**
8	11	12.76 ± 7.69	12	18.301 ± 4.919	**0.050**
10	11	14.95 ± 7.11	12	23.076 ± 6.535	**0.009**
12	9	12.96 ± 6.28	12	19.818 ± 5.636	**0.017**
Average speed (cm/s)					
2	9	12.65 ± 2.79	7	11.44 ± 4.15	0.496
3	11	10.96 ± 3.94	12	12.61 ± 2.99	0.268
4	8	10.76 ± 4.27	10	11.38 ± 2.62	0.705
5	12	7.26 ± 2.76	12	6.30 ± 3.19	0.442
6	11	6.80 ± 2.39	10	5.69 ± 2.38	0.297
7	8	7.01 ± 3.43	12	5.27 ± 2.57	0.211
8	11	6.61 ± 2.79	12	6.68 ± 4.12	0.959
10	11	6.49 ± 2.18	12	6.93 ± 2.45	0.651
12	9	5.77 ± 1.49	12	6.62 ± 3.03	0.445

Note: The statistics presented are from the *t*-test, comparing groups at each time point, with *p*-values < 0.05 in bold.

**Table 3 ijms-26-03657-t003:** Descriptive and statistical analysis of novel object recognition data for Alzheimer’s disease and control groups: comparison of recognition index and average speed.

Time (Months)	Groups				Statistics *p*-Value
	Control		Alzheimer Disease		
	N	Mean ± SD	N	Mean ± SD	
Recognition index					
2	12	68.89 ± 18.99	10	72.93 ± 24.16	0.665
3	12	75.20 ± 24.76	11	83.49 ± 25.11	0.435
4	9	70.87 ± 29.37	9	36.30 ± 16.30	**0.007**
5	11	75.53 ± 18.60	10	38.18 ± 22.78	**<** **0** **.001**
6	8	76.50 ± 23.38	9	58.10 ± 13.22	0.061
7	12	72.18 ± 18.57	9	53.96 ± 10.21	**0.016**
8	7	78.25 ± 15.82	6	61.36 ± 15.00	0.075
10	6	85.50 ± 20.01	9	52.43 ± 23.42	**0.014**
12	3	78.42 ± 15.78	8	51.13 ± 12.17	**0.013**
Average speed (cm/s)					
2	12	12.14 ± 3.49	10	8.27 ± 2.08	**0.006**
3	12	11.27 ± 2.42	11	8.42 ± 2.84	**0.017**
4	9	9.31 ± 3.39	9	7.14 ± 1.75	0.107
5	11	11.18 ± 3.92	10	9.04 ± 2.72	0.167
6	8	10.15 ± 2.15	9	10.79 ± 1.91	0.529
7	12	5.78 ± 1.41	9	7.41 ± 2.36	0.063
8	7	7.87 ± 0.84	6	7.02 ± 2.82	0.459
10	6	5.54 ± 1.53	9	9.69 ± 2.38	**0.002**
12	3	6.69 ± 1.71	8	6.93 ± 2.47	0.882

Note: The statistics presented are from the *t*-test, comparing groups at each time point, with *p*-values < 0.05 in bold.

**Table 4 ijms-26-03657-t004:** Descriptive and statistical analysis of oligoelements based on the concentrations of copper (Cu), selenium (Se), and zinc (Zn) over 12 months in the Alzheimer’s disease (AD) group and control group.

Time (Months)	Control (µg/mL)	AD (µg/mL)	*t*-Student
Copper (Cu)	Mean ± SD	Mean ± SD	*p*-Value
2	1.30 ± 0.28	1.23 ± 0.14	0.440
3	1.28 ± 0.29	1.63 ± 0.38	**0.017**
4	1.26 ± 0.23	1.59 ± 0.30	**0.007**
5	1.23 ± 0.25	1.48 ± 0.34	0.054
6	1.21 ± 0.22	1.45 ± 0.24	**0.022**
7	1.13 ± 0.18	1.55 ± 0.15	**<0.001**
8	1.28 ± 0.27	1.44 ± 0.10	0.071
9	1.22 ± 0.33	1.47 ± 0.19	**0.034**
10	1.32 ± 0.11	1.54 ± 0.19	**0.002**
11	1.30 ± 0.09	1.57 ± 0.18	**<0.001**
12	1.20 ± 0.24	1.62 ± 0.173	**<0.001**
Selenium (Se)			
2	1.21 ± 0.31	1.22 ± 0.21	0.895
3	1.28 ± 0.27	0.98 ± 0.19	**0.006**
4	1.16 ± 0.17	0.88 ± 0.19	**0.001**
5	1.03 ± 0.33	0.73 ± 0.32	**0.033**
6	0.98 ± 0.29	0.53 ± 0.29	**0.001**
7	0.97 ± 0.21	0.54 ± 0.40	**0.004**
8	0.93 ± 0.26	0.56 ± 0.17	**<0.001**
9	0.96 ± 0.50	0.42 ± 0.26	**0.003**
10	1.090 ± 0.528	0.563 ± 0.57	**0.030**
11	0.763 ± 0.235	0.314 ± 0.152	**<0.001**
12	0.621 ± 0.281	0.393 ± 0.118	**0.016**
Zinc (Zn)			
2	3.82 ± 0.38	3.83 ± 3.83	0.890
3	3.75 ± 0.09	3.53 ± 3.53	**<0.001**
4	3.41 ± 0.08	3.47 ± 3.47	0.080
5	3.75 ± 0.10	3.01 ± 3.01	**<0.001**
6	3.33 ± 0.25	3.06 ± 3.06	**<0.001**
7	3.54 ± 0.25	2.89 ± 2.89	**<0.001**
8	3.25 ± 0.39	2.64 ± 2.64	**<0.001**
9	3.43 ± 0.44	2.25 ± 2.25	**<0.001**
10	2.93 ± 0.27	1.81 ± 1.81	**<0.001**
11	2.65 ± 0.20	1.86 ± 1.87	**<0.001**
12	2.59 ± 0.14	1.55 ± 1.55	**<0.001**

Note: The statistics presented are from the *t*-test, comparing groups at each time point, with *p*-values < 0.05 in bold.

## Data Availability

The original contributions presented in the study are included in the article. Further inquiries can be directed to the corresponding author.

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
