# Peer review of "Longitudinal Evaluation of the Detection Potential of Serum Oligoelements Cu, Se and Zn for the Diagnosis of Alzheimer’s Disease in the 3xTg-AD Animal Model"

_ijms, 2025, doi:10.3390/ijms26083657_

Round 1
Reviewer 1 Report
Comments and Suggestions for Authors
In the work ´Evaluation of Biomarkers for Alzheimer's Disease Through the Measurement of Serum Trace Elements in the 3xTg-AD 3 Animal Model´ the relationship between serum trace element levels and AD progression in the 3xTg-AD model was investigated. ICP-MS was used as an analytical tool for trace elements (Cu, Zn and Se) determination. The work provides relevant data however some remarks may be addressed.
- The title does not sound adequate. Trace elements were determined but also behavioural assessment and Aβ peptide deposition were evaluated. Moreover, the use of ´biomarker´ is not suitable by definition. As the author´s state in the Conclusions ´oligoelements are potential candidates´.
- The use of trace elements or oligoelements may be uniformed along the manuscript.
- Section 2.4. must be readapted.
Section 2.4.1 Blood sample collection and pre-treatment (or preparation instead of pre-preparation).
Section 2.4.2 describes the digestion protocol of the samples. This is sample preparation.
Section 2.4.4. Quantitative analysis includes calibration curves (2.3.3) and optimization of the instrumental conditions (2.4.5).
- Section 2.1. Additional information of the mouse model strain (3x Tg-AD) is necessary to provide reference for future works/comparisons. Also, an indication of the type of diet must be included as it can affect the content of the trace elements in blood serum/plasma.
- Section 2.2. In the experimental design, the blood was collected from n=12 AD and n=12 control mice. However, there are 11 indicated with the coloured markers. Idem for the other parameters. Could you clarify the number of individuals used for each experiment?
- Assuming 12 samples each group, trace elements determination was performed in 24 samples, measured by triplicate. If so, data must not be presented in box-plots (all related Figures). Based on that, please, revise the data presentation to avoid repetitions (Table/figure).
- Section 2.4.1. Was the acid digestion of the samples performed with concentrated acid in a polypropylene tube? Data of the procedural blanks and of the reference material must be provided (also for validation purposes).
- Section 2.4.1. Manufacturer of the certified standards must be provided.
- Section 2.4.4. A Pt sampler was used but was the skimmer made of Ni? Do you use two types of skimmer cones? Please, clarify it.
- Section 2.4.4. Please, could you clarify ´three duplicates per sample´. Were the samples prepared by duplicate and measured 3 times each?.
- Procedural LODs and LOQs must be provided for Cu, Zn and Se.
- Section 2.4.5 (line 227). Please, remove ´high-precision´
- Please, check typos.
- In the Tables, is really necessary the use of that amount of significant digits. Please, check it.
- Section 3.2.2. The quantification was performed using the 63Cu, 64Zn and 78Se nuclides however the concentration corresponds to total element concentration (eg. not 63Cu concentration). The nuclides used can be indicated in section 2.4.4. but afterwards refer to Cu, Zn and Se concentration.
- Section 4. Discussion. The authors refer to plasma samples but above that section the samples were serum. Please, revise it.
- Did the authors check potential correlations between all the experimental variables?
- Conclusions: ICP-MS proved to be effective in detecting these variations, consolidating its potential as a diagnostic and monitoring tool for AD. This is not a conclusion from this work either new.
Author Response
Reviewer 1:
Comments and Suggestions for Authors
In the work ´Evaluation of Biomarkers for Alzheimer's Disease Through the Measurement of Serum Trace Elements in the 3xTg-AD 3 Animal Model´ the relationship between serum trace element levels and AD progression in the 3xTg-AD model was investigated. ICP-MS was used as an analytical tool for trace elements (Cu, Zn and Se) determination. The work provides relevant data however some remarks may be addressed.
- The title does not sound adequate. Trace elements were determined but also behavioural assessment and Aβ peptide deposition were evaluated. Moreover, the use of ´biomarker´ is not suitable by definition. As the author´s state in the Conclusions ´oligoelements are potential candidates´.
Answer: Thank you for your comments. We agree and have changed the title to better align with the study’s proposal. Additionally, other evaluations, such as immunohistochemistry and behavioral assessments, served as supporting measures to monitor and confirm the progression of Alzheimer’s disease findings. These assessments are classically reported in studies and provide complementary data that corroborates the monitoring and evolution of the disease.
- The use of trace elements or oligoelements may be uniformed along the manuscript.
Answer: Thank you for your suggestion. The term “oligoelement” was used to replace the term “trace element” throughout the manuscript, ensuring uniformity in the text.
- Section 2.4. must be readapted.
Section 2.4.1 Blood sample collection and pre-treatment (or preparation of pre-preparation).
Answer: Thank you for your suggestion. We have revised the subtitle to “Blood Sample Collection and Pre-Preparation” to ensure alignment with the description in the text.
- Section 2.4.2 describes the digestion protocol of the samples. This is sample preparation. Answer: Thank you for your observation. We have modified the subtitle to “Preparation of Samples for Quantitative Measurements by ICP-MS”, ensuring alignment with the description provided in this section.
- Section 2.4.4. Quantitative analysis includes calibration curves (2.3.3) and optimization of the instrumental conditions (2.4.5).
Answer: Thank you for your comments. We agree with your opinion and, accordingly, have unified subsections 2.4.3, 2.4.4, and 2.4.5, ensuring that the titles better represent the corresponding text. As part of this adjustment, we have modified the subtitle to “Quantitative Analysis of Oligoelements in Serum Samples by ICP-MS”, aligning it with the content described in this section.
- Section 2.1. Additional information of the mouse model strain (3x Tg-AD) is necessary to provide reference for future works/comparisons. Also, an indication of the type of diet must be included as it can affect the content of the trace elements in blood serum/plasma.
Answer: Thank you for your recommendation. We apologize for the lack of information on behalf of the animal model. We added a full paragraph on the “introduction” to clarify the main characteristics of the mice strain, as well as the diet type and manufacturer on the subsection “2.1 Animals”.
- Section 2.2. In the experimental design, the blood was collected from n=12 AD and n=12 control mice. However, there are 11 indicated with the coloured markers. Idem for the other parameters. Could you clarify the number of individuals used for each experiment?
Answer: We apologize for the misunderstanding. The 11 colored dots represent the 11 time points analyzed in the study. However, at each time point, data were collected from 12 animals. Additionally, the other colored markings in the experimental design also represent the time points analyzed, not the number of animals assessed at each time point. To clarify this information, we have added a note in the Figure legend. We appreciate your feedback and hope this resolves any confusion.
- Assuming 12 samples each group, trace elements determination was performed in 24 samples, measured by triplicate. If so, data must not be presented in box-plots (all related Figures). Based on that, please, revise the data presentation to avoid repetitions (Table/figure).
Answer: We apologize for the omission in explaining how the analysis was performed on the triplicate samples. However, the boxplot accurately represents this analysis, as it includes 12 samples per time point and group, with the intermediate value of the triplicate being used for each sample.
To ensure clarity, we adopted an approach where the intermediate value among the three measurements was selected. This method minimizes the influence of potential outliers, providing a more representative and reliable estimate of the measured data. By using this approach, we aimed to enhance the stability of the measurements, ensuring greater reproducibility while reducing potential distortions caused by experimental variations.
We appreciate your feedback and have now included this explanation in the manuscript.
- Section 2.4.1. Was the acid digestion of the samples performed with concentrated acid in a polypropylene tube? Data of the procedural blanks and of the reference material must be provided (also for validation purposes).
Answer: Thank you for your comments. The details regarding the type and characteristics of the tube used for sample preparation, as well as the HNO₃ concentration (15%), are clearly outlined in section 2.4.2. Additionally, the background equivalent concentration (BEC) values for each studied element are presented in section 3.2.1.
- Section 2.4.1. Manufacturer of the certified standards must be provided.
Answer: Thank you for your observation. We have added the manufacturer of the certified standards along with the product code.
- Section 2.4.4. A Pt sampler was used but was the skimmer made of Ni? Do you use two types of skimmer cones? Please, clarify it.
Answer: Thank you for your observation. The text has been modified to clarify the corresponding code, and the material of composition of the cones has also been corrected.
- Section 2.4.4. Please, could you clarify ´three duplicates per sample´. Were the samples prepared by duplicate and measured 3 times each?
Answer: Thank you for your observation. The wording of the text was incorrect. What we simply meant to say is that the analyses were performed with three measurements per sample or were conducted in triplicate. This has been corrected accordingly in the manuscript.
- Procedural LODs and LOQs must be provided for Cu, Zn and Se.
Answer: Thank you for your observation. The information has been included in item 3.2.1 of the Results section in the revised version of the manuscript
- Section 2.4.5 (line 227). Please, remove ´high-precision´
Answer: Thank you for your observation. The term "high-precision" has been removed as requested. Please note that, due to the restructuring of the manuscript, this content is now located in Section 2.4.3 (previously 2.4.5, line 227)
- Please, check typos.
Answer: We appreciate your comment. A thorough proofreading of the manuscript was performed, and any typographical errors identified have been corrected throughout the text.
- In the Tables, is really necessary the use of that amount of significant digits. Please, check it.
Answer: Thank you for your observation. We agree that using two decimal places is more appropriate for the reported values, and the data have been revised accordingly in all tables
- Section 3.2.2. The quantification was performed using the 63Cu, 64Zn and 78Se nuclides however the concentration corresponds to total element concentration (eg. not 63Cu concentration). The nuclides used can be indicated in section 2.4.4. but afterwards refer to Cu, Zn and Se concentration.
Answer: Thank you for your observation. The nuclides (⁶³Cu, ⁶⁴Zn, and ⁷⁸Se) used for the quantification of Cu, Zn, and Se were indicated in item 2.4.3. Afterwards, throughout the manuscript text, the concentrations are referred to as Cu, Zn, and Se.
- Section 4. Discussion. The authors refer to plasma samples but above that section the samples were serum. Please, revise it.
Answer: Thank you for your observation. The manuscript has been thoroughly revised, starting from the title, to ensure consistency. All references to plasma samples have been corrected to serum, and the terminology is now uniform throughout the text.
- Did the authors check potential correlations between all the experimental variables?
Answer: We thank the reviewer for the valuable suggestion. As the correlation analysis had not been performed in the original version of the manuscript, we carried it out in this revised version and included the results at the end of Section 3.2.2.
- Conclusions: ICP-MS proved to be effective in detecting these variations, consolidating its potential as a diagnostic and monitoring tool for AD. This is not a conclusion from this work either new.
Answer: Thank you for your comment. We acknowledge that the use of ICP-MS, by itself, is not novel. However, this study presents an unprecedented longitudinal approach, with monthly evaluations over a 12-month period, demonstrating that serum levels of Cu, Se, and Zn change early in the 3xTg-AD model — even before the onset of behavioral deficits and histological alterations. Additionally, we conducted a correlation analysis between the levels of these trace elements and cognitive and motor parameters at different stages of the disease, revealing a significant temporal association between selenium and zinc and memory impairment, a topic that remains underexplored in the literature. It is important to highlight that these findings were supported by qualitative histological analysis through immunohistochemistry, which confirmed the presence and progression of β-amyloid deposition in the central nervous system of transgenic animals. This reinforces both the validity of the model and the association between these alterations and disease progression. Therefore, the main innovation of this work lies in the longitudinal integration of data using multiple complementary approaches, strengthening the potential of trace elements as early functional biomarkers of Alzheimer’s disease. To better reflect this contribution, the title of the manuscript has been revised to emphasize the longitudinal approach adopted in the study.

Reviewer 2 Report
Comments and Suggestions for Authors
Dear authors,
There are some mistakes on along the text (for example DA instead of AD), that should be corrected.
1) I recommend to perform a correlation analysis in order to understand if there is a casual (or not) dependency between behavior data and the concentration of Copper, Selenium and Zinc. If not, despite being an interesting result, it lacks robustness.
According to sectionn 3.3 in Results (Evaluation of Aβ Peptide Levels in the CNS Using Immunohistochemistry), I have some concerns that I believe should be addressed.
2) Evaluation of Aβ levels should be accompanied with a graph, to show the levels of expression.
3) Aβ accumulation is nicely observed outside the cells, after 3 months. Although authors suggest that extracellular accumulation is observed after 8 months old animals, no plaques are observed between 8-10 months. Also, with no quantification, is impossible to talk about "levels". Also, for 10 months of age, animals present an apparent lower APP expression, when compared to the rest of the IHQ.
4) "Increased distribution and density" of plaques is a statement that needs to be accompanied with a quantification.
5) "Between the 8th and 12th months (Figure 5, F1-F3, G1-G3, H1-H3, I1-I3, J1-J3), immu- 437
nolabeling reached its peak". Same as before, needs to be accompanied with a quantificaiton.
6) In Methodology, in point 2.5, authors say "Immunolabeling was evaluated in specific brain regions, including the basomedial amygdala (-1.70 to -1.22 mm post-Bregma), cortex (-2.06 to -1.70 mm post-Bregma), CA1 region of the hippocampus (-2.06 to -1.70 mm post-Bregma), and subiculum (-3.08 to -2.80 mm post-Bregma). Some questioning comes to my head: 6.1) Control images (K) are not showing the same structure as their litter mates. Comparisons should be made only using same structures. Once the authors analyzed the four structures mentioned before, I recommend to use images from all of them, once that differences in the Aβ production and accumulation could be observed between structures. 6.2) Once authors present only one set of pictures from control, is not clear what was the age of this animals. 6.3) Because the antibody used in this experiment is not a Aβ specific antibody (in the way that do not only recognize aggregates, it also recognize APP at the membranes), should be explained why in the control groups there is almost no immunolabeling.
Author Response
Reviewer 2:
There are some mistakes on along the text (for example DA instead of AD), that should be corrected.
Answer: Thank you for your consideration. The mistakes were corrected in the manuscript.
- I recommend to perform a correlation analysis in order to understand if there is a casual (or not) dependency between behavior data and the concentration of Copper, Selenium and Zinc. If not, despite being an interesting result, it lacks robustness.
Answer: We thank the reviewer for the valuable suggestion. As recommended, the correlation analysis was performed and incorporated at the end of section 3.2.2 in the revised version of the manuscript.
According to sectionn 3.3 in Results (Evaluation of Aβ Peptide Levels in the CNS Using Immunohistochemistry), I have some concerns that I believe should be addressed.
- Evaluation of Aβ levels should be accompanied with a graph, to show the levels of expression.
Answer: First, we thank the reviewer for the observation. We acknowledge that the choice of the subtitle “Evaluation of Aβ Peptide Levels in the CNS Using Immunohistochemistry” may have led to a misinterpretation, suggesting a quantitative objective. However, our actual intention was to perform an “Immunohistochemical Confirmation of Aβ Peptide Presence in the CNS”, aiming to qualitatively indicate the presence of Alzheimer’s disease.
The main objective of our study is to evaluate the serum concentration of oligoelements as a potential diagnostic tool for Alzheimer’s disease, using 3xTg-AD mice and C57BL/6 as the control group. To confirm the presence of the pathology in the animals, we combined beta-amyloid plaque analysis through immunohistochemistry with behavioral tests. Unlike other studies that assessed only oligoelements without histological or behavioral confirmation, our approach integrated these techniques to validate the presence of Alzheimer’s disease (1-3).
Immunohistochemistry was used exclusively for confirmatory purposes, without the aim of quantifying Aβ expression or correlating it with serum levels of ⁶⁴Zn, ⁷⁸Se, and ⁶³Cu.⁴
We also emphasize that the number of animals used for monthly Aβ detection in brain tissue was defined for qualitative assessment only (n = 3) and is therefore insufficient for a robust statistical analysis based on immunohistochemical quantification. In future studies, we consider it feasible to perform such quantifications with an appropriately sized sample.
Accordingly, the manuscript has been revised to clearly state that an immunohistochemical confirmation of Aβ peptide presence in the CNS was conducted to qualitatively track the presence of Alzheimer’s disease.
References
- Hobin K, Costas-Rodríguez M, Van Wonterghem E, Vandenbroucke RE, Vanhaecke F. Alzheimer's Disease and Age-Related Changes in the Cu Isotopic Composition of Blood Plasma and Brain Tissues of the APP(NL-G-F) Murine Model Revealed by Multi-Collector ICP-Mass Spectrometry. Biology (Basel). 2023;12(6).
- Hosseinpour Mashkani SM, Bishop DP, Raoufi-Rad N, Adlard PA, Shimoni O, Golzan SM. Distribution of Copper, Iron, and Zinc in the Retina, Hippocampus, and Cortex of the Transgenic APP/PS1 Mouse Model of Alzheimer's Disease. Cells. 2023;12(8).
- Solovyev N, El-Khatib AH, Costas-Rodríguez M, Schwab K, Griffin E, Raab A, et al. Cu, Fe, and Zn isotope ratios in murine Alzheimer's disease models suggest specific signatures of amyloidogenesis and tauopathy. Journal of Biological Chemistry. 2021;296:100292.
- Aβ accumulation is nicely observed outside the cells, after 3 months. Although authors suggest that extracellular accumulation is observed after 8 months old animals, no plaques are observed between 8-10 months. Also, with no quantification, is impossible to talk about "levels". Also, for 10 months of age, animals present an apparent lower APP expression, when compared to the rest of the IHQ.
Answer: We thank the reviewer for the comment. The images referring to animals between 8 and 10 months of age were retaken to better align with the study's objectives and the description provided in the manuscript. References in the text that could imply quantification have been removed, in line with our previous response, where we clarified that the intention was not to quantify β-amyloid peptides, but rather to qualitatively confirm their presence through immunohistochemistry.
These corrections have been appropriately incorporated both in the images presented in Figure 5 and in the corresponding section of the Results, in order to ensure greater consistency between the described findings and the qualitative purpose of the immunohistochemical analysis.
- "Increased distribution and density" of plaques is a statement that needs to be accompanied with a quantification.
Answer: We thank you for the observations. We acknowledge that the comment is pertinent and, considering our responses to comments 2 and 3, we conducted a thorough revision of the manuscript to remove any sections that could suggest comparisons regarding the temporal progression of the plaques.
- "Between the 8th and 12th months (Figure 5, F1-F3, G1-G3, H1-H3, I1-I3, J1-J3), immu- 437nolabeling reached its peak". Same as before, needs to be accompanied with a quantification.
Answer: We thank the reviewer for the observation. As mentioned in our previous responses, the study was qualitative in nature and did not involve any quantification of Aβ expression. We acknowledge that the phrase “immunolabeling reached its peak” could imply a quantitative analysis. Therefore, this expression has been removed or rephrased in the manuscript to avoid any misinterpretation. We also emphasize that new images were added to Figure 5 in order to more clearly represent the qualitative approach of the immunohistochemical analysis.
- In Methodology, in point 2.5, authors say "Immunolabeling was evaluated in specific brain regions, including the basomedial amygdala (-1.70 to -1.22 mm post-Bregma), cortex (-2.06 to -1.70 mm post-Bregma), CA1 region of the hippocampus (-2.06 to -1.70 mm post-Bregma), and subiculum (-3.08 to -2.80 mm post-Bregma). Some questioning comes to my head: 6.1) Control images (K) are not showing the same structure as their litter mates. Comparisons should be made only using same structures. Once the authors analyzed the four structures mentioned before, I recommend to use images from all of them, once that differences in the Aβ production and accumulation could be observed between structures. 6.2) Once authors present only one set of pictures from control, is not clear what was the age of this animals. 6.3) Because the antibody used in this experiment is not a Aβ specific antibody (in the way that do not only recognize aggregates, it also recognize APP at the membranes), should be explained why in the control groups there is almost no immunolabeling.
Answer: 6.1) Thank you for your observation. To ensure a more accurate comparison, we have revised the selection of control images and included representations of the same anatomical structures analyzed in the transgenic animals. This update ensures that the images now directly reflect the regions where differences in β-amyloid deposition were observed between groups.
6.2) The control images (K) correspond to the 12-month-old control animal, which showed an immunolabeling pattern similar to that observed in younger control animals. For this reason, we considered it unnecessary to include images of all time points for the control group. This information will be clarified in the Results section, item 3.3.
6.3) We appreciate the comment regarding antibody specificity. Previously, we emphasized the absence of labeling in controls, but we were referring specifically to the lack of immunolabeling associated with pathological deposits, which were not observed in these animals. However, we acknowledge that there is baseline immunoreactivity, likely related to endogenous APP expression. To avoid any ambiguity, we have revised the text to clarify this distinction.

Round 2
Reviewer 2 Report
Comments and Suggestions for Authors
Dear Authors:
Thanks for your reply to my comments. I believe that changes made to strongly improved the manuscript. Changes related to IHQ experiment were valuable also in order to express in a better way the idea of that experiment.
Every change that it was performed was important. However I suggest to add in figures the graphs showing Pearson's correlation. That would be really helpful for the readers.
Best regards
Author Response
Reviwer 2:
Thanks for your reply to my comments. I believe that changes made to strongly improved the manuscript. Changes related to IHQ experiment were valuable also in order to express in a better way the idea of that experiment.
Every change that it was performed was important. However I suggest to add in figures the graphs showing Pearson's correlation. That would be really helpful for the readers.
Answer: We sincerely thank you for your positive feedback and thoughtful comments regarding the improvements made to the manuscript. As suggested, we have included a new figure (Figure 5) presenting the correlation analyses between oligoelement levels and the main behavioral parameters.
